# Factors impacting employee turnover intentions among professionals in Sri Lankan startups

**Lakshmi Kanchana[1,2], Ruwan Jayathilaka[3]***

**1** SLIIT Business School, Sri Lanka Institute of Information Technology, Malabe, Sri Lanka, **2** Ceyentra Technologies, Panadura, Sri Lanka, **3** Department of Information Management, SLIIT Business School, Sri Lanka Institute of Information Technology, Malabe, Sri Lanka

* ruwan.j@sliit.lk

**Data Availability Statement:** All relevant data are within the paper and its Supporting Information files (S2 Appendix. Data File).

**Funding:** The authors received no specific funding for this work.

## Abstract

Employee turnover is one of the topical issues worldwide. The impact of factors affecting employee turnover varies occasionally and new factors are considered. Many countries have examined various factors that affect employee turnover. The main objective of this research is to consider psychographics and socio-demographic factors in one study and analyse the impact on employee turnover. A Probit regression model through the stepwise technique was used to analyse the collected data. Using ventures in Sri Lanka as a case study, this study demonstrates that employee turnover occurs in different stages and independent factors impact differently in each stage. The study population was professionals who have been a key part of Sri Lankan startups, which involved 230 respondents. Data analysis was performed through a forward stepwise technique through STATA. The results verified that job satisfaction and co-worker support negatively impact employee turnover, whereas leader member exchange positively impacts employee turnover. This study also proved a significant positive relationship between male employees in their thirties and high employee turnover. This study's findings help to identify the areas management should focus on to minimise employee turnover to retain experienced and skilled employees.

## Introduction

Having the right combination of human resources/employees can assist firms to be effective in driving change, boosting business performance, as well as to achieving and sustaining a competitive edge. Companies need to give high priority to employee development and predict employee behaviour [1]. Organisations spend more time and take much effort to identify the good fit employees for the company. Companies invest in many ways for employees, as they are one of the organisation's valuable assets [2]. Organisations conduct workshops for employees, buy online tutorials, evaluate employee performance, and provide feedback to them, which are some common types of investments in human resources. These processes sharpen employees' skills and capabilities, directly affecting the organisation's success. However, some organisations are weak in strategy adoption while not focusing constantly on these processes

**Competing interests:** The authors have declared that no competing interests exist.

or employee voice. As such, these employees suddenly quit the company resulting in increased employee turnover. The issue of employee turnover is considered as one of the global obstacles for organisations worldwide, which directly and adversely affects strategic plans and opportunities of gaining competitive advantages [3]. As such, this issue can have massive effects on a company's performance, especially for new businesses and startups. Therefore, it is essential to identify the factors that affect employee retention, which is also a topical issue worldwide. This type of approach enables businesses to achieve its strategic goals while retaining satisfied and skilful employees.

Many variables influence employee turnover intentions [4–6]. Previous studies imply that job satisfaction, work-life balance, trust, and management support are the critical factors that impact employee retention [7–9]. Further, promoting employee well-being leads to decrease employee turnover [10]. Providing psychological and social support through counselling promotes the quality of work-life [11]. With time, newly considered factors such as leader member exchange, workplace culture, happiness, joy in the workplace, career management, innovative work behaviour and employee delight are equally important and have been identified. As such, it is important to focus on these factors and build relationships between employees and the organisation.

Firm performance reflects the ability of an organisation to use its human resources and other material resources to achieve its goals and objectives. Firm performance belongs to the economic category, and it should consider the use of business means efficiently during the production and consumption process [12]. Employee retention is defined as encouraging employees to remain in the organisation for a long period or the organisation's ability to minimised employee turnover [13]. Turnover intention is the intention of the employee to change the job or organisation voluntarily [14].

Sri Lankan business firms were chosen as a case study to examine this resarch problem. In Sri Lanka, over 1 million (Mn) businesses operate. By 2018, 10,510 new businesses had been registered in Sri Lanka. Among these companies, startup companies play a key role in the Sri Lankan economy. Startups come up with radical innovations and changes, and these disrupt the existing market with new products and services. Furthermore, Sri Lanka has a middle rank of ease of doing business. With these favourable conditions and educational and family backgrounds, many people like to apply their new idea and fill the market gap. The new generation in Sri Lanka are interested/are keen on innovations at work and being a part of unique products or services. Currently, most startups are technology-driven and do not have geographical limitations.

Startups are expanding day by day. These businesses are in different stages as ideation, traction, break-even, profit, scaling and stable. According to the "Sri Lanka Startup Report 2019" issued by PricewaterhouseCoopers (PWC), "55% of startups responded are in the growing revenue or expansion stage, 29% of respondents reported an annual revenue of more than Sri Lankan Rupees (LKR) 10 Mn, 40% are still in the less than LKR 1 Mn revenue category and 61% of respondents reported being profitable". In this setting, employee turnover can be a setback for most startups yet to reach business stability.

Most startups are relatively new. According to (PWC) [15], 36% of the businesses have operated for less than a year, 44% have been in operation for 1–3 years and only 8% have operated for more than five years. These are still growing and in the early stages of executing their strategies. In this situation, most companies are willing to expand their staff strength. PricewaterhouseCoopers [15] evidenced that 82% of companies were willing to do so in the next year.

Studies conducted in Asian countries on this subject are assumably similar to the situation of Sri Lanka [4, 5, 16]. This study aims to create a model with critical and newly identified independent factors (job satisfaction, work-life balance, happiness, management support,

career management, innovative work behaviour, leader member exchange, and co-worker support) influencing employee turnover in Sri Lankan startups.

Based on their knowledge and the existing literature, authors have considered widely used factors to investigate the employee turnover issue. Therefore, job satisfaction, happiness, work-life balance, career management, management support, innovative work behaviour, leader member exchange and co-worker support were selected based on previous literature findings [4–6, 8, 17–19]. As in the previous papers and along with the current study's results, authors identified both positive and negative impacts on employee turnover among Sri Lankan startups.

This study aims to analyse the impact of job satisfaction, happiness, work-life balance, career management, management support, innovative work behaviour, leader member exchange, and co-worker support on employee turnover in startups in Sri Lanka. The present study's scientific value can be elaborated by comparing it with previous studies. This study's contribution can be explained in five ways. Firstly, the most critical and newly considered factors were identified together with the support of past literature. Secondly, the present study was classified into different levels of employee turnover. As such, by considering the various levels, the micro-level changes and probabilities of the impact on employee turnover can be better identified. Further, this study helps to reduce the methodological gap. Thirdly, the Sri Lankan context has been selected as the case study. This is because, to the best of the authors' knowledge, there was no previous research done by local researchers that includes all the widely measured variables investigating the combined effect on employee turnover. Fourthly, the analysis results can be used to identify the strengths and weaknesses of startups in Sri Lanka. Finally, this study identifies the challenges faced by startups and identifies how policy modifications can strengthen the startup ecosystem.

The upcoming sections of this paper are structured as follows. Section 2 discusses the literature review, and section 3 explains data and methodology, Section 4 contains results and discussion highlighting how the research objectives are achieved. Section 5 marks the conclusion, with implications, research limitations and future research directions.

## Literature review

As employee turnover is one of the most critical indicators for an organisation, many studies have been conducted on this topic with dissimilar demographical and geographical samples. The existing literature adds theoretical or methodological improvements to this topic. Accoridngly, this study included most variables that significantly impact employee turnover, summarising the independent variables that affect employee retention.

This study is based on the initially defined 47 journal articles through advanced filtration. Reputed journal databases, such as Emerald insight, Science Direct, Taylor & Francis, SAGE journals, ResearchGate, Sabinet, IEEE Xplore and Google Scholar were referred. Fig 1 below describes the literature search flow. Thirteen articles were excluded due to overlapping, insufficient information and irrelevant to the topic. The selected articles have been sorted according to the independent variables.

S4 Appendix contains the literature summary of the above presented literature search flow diagram. The following sections present the details of each category.

## Job satisfaction

Job satisfaction refers to the employee's positive emotions, feeling and attitudes on the job and workplace. Positive emotional experiences directly affect higher job satisfaction [7]. Kim, Knutson [7] found that satisfaction significantly affects employee turnover regardless of the

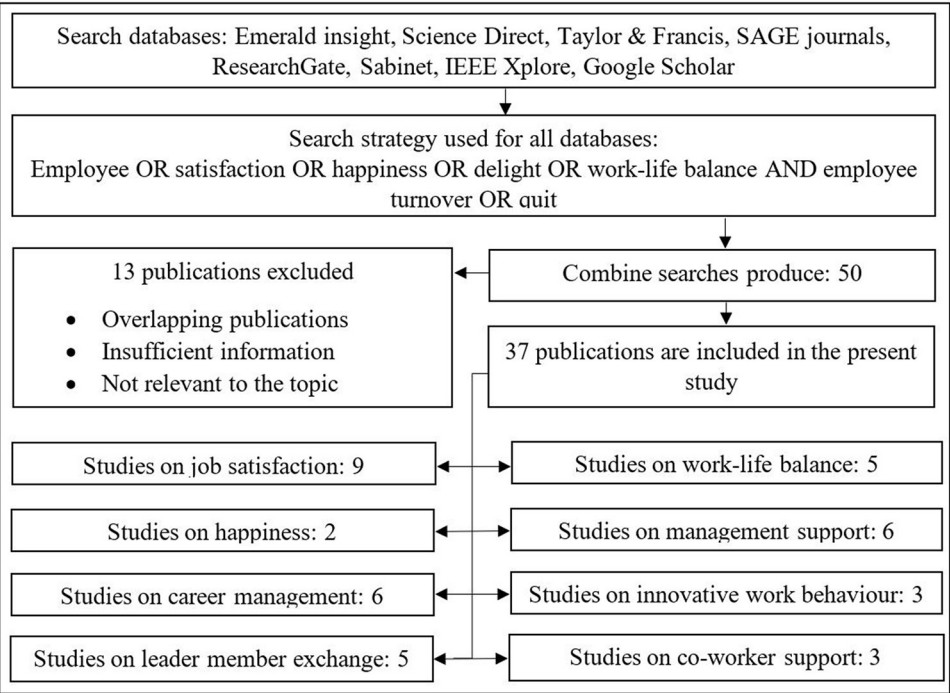

**Fig 1. Literature search flow diagram.** Source: Based on authors' observations.

generation of the employee. Gen Y employees do not easily build loyalty toward the organisation unlike older employees. Turnover intentions seem significantly higher in new generations compared to older generations. New generations are impatient with their organisation and older generations are more patient with it. However, even the new generation of employees tends to stay in their organisation if their level of satisfaction is acceptable. They found that newer the generation of employee, satisfaction level and loyalty is lower than the older generation. This shows that employee turnover is higher in newer generations. Feedback obtained from most employees in generations Y and Z in startups supports this finding.

Da Camara, Dulewicz [20] found that organisational emotional intelligence has a larger effect on employee satisfaction. Further, this study has discovered that organisational emotional intelligence helped improve job satisfaction and commitment, which reduced turnover intentions significantly. However, organisational commitment and satisfaction describe only 19% of the total intention to leave. Moreover, the descriptive statistics found a high level of job satisfaction and the intention to leave was at the mid or average level of the scale. Camara further stated that job satisfaction clearly implies the feeling about their job. But some research findings can be contradictory. Some employees are fully satisfied with the job and still want to leave the organisation for various reasons. However, this research focused only on charity workers. As such, it is important to gather many indicators that affect employee turnover and thereafter, one can analyse the real situation and generalise the findings.

Satisfaction also depends on the number of employees at the same level. When it gets higher, job satisfaction increases and reduces the intention to leave [8]. This study found that female employees are more satisfied with their jobs, while older employees are more likely to leave the organisation. However, this study focused only on online-level employees and supervisors.

Oosthuizen, Coetze and Munro studied the relationship between job satisfaction and turnover intention in the IT industry. Oosthuizen, Coetzee [6] revealed that job satisfaction

significantly predicted employee turnover. The study also found that the work-home life balance has a major effect on job satisfaction. Predicting turnover intention based on overall work-life balance is a tough task. The findings further proved that white employees show less job satisfaction compared to black employees. However, they didn't observe any significant interaction between overall work-life balance and job satisfaction in predicting employee turnover intention. With these results, this indicator must be examined further.

Considering the Asian context, Pakistan IT professionals' turnover intentions were studied in a similar research [21]. Recruitment & section, team & management support, performance & career management, salary & compensation, employee commitment, job security, recognition, organisational demographics, and personal demographics have an effect on job satisfaction. However, this study suggested adding more factors, such as work-life balance and employee engagement, which may significantly impact employee retention. This means that human resource management has a significant influence on job satisfaction.

The study by Zeffane and Bani Melhem [22] investigated the turnover intention of public and private sector employees in the United Arab Emirates. Here, the researchers revealed that government employees are more satisfied with their job and are most unlikely to leave than private sector employees. The turnover intentions of private sector employees are not significantly affected by job satisfaction, whereas the public sector is almost affected by it. Kaur and Randhawa [16] examined the turnover intention of Indian private school teachers. It revealed that job satisfaction has a direct link with the civil status of the teachers, explaining that married teachers tend to have less job satisfaction. However, for unmarried teachers, there is more intention to leave organisations. Supervisor's influence had indirect impacts on turnover intentions. However, this research limited the sample to private school female teachers. Here, the study highlighted the importance of having more influencing variables on employee retention and recommended considering these for a comprehensive analysis. Only then the model can be near to the real situation.

Thomas A. Wright [2] discovered that the employee's well-being moderates the relationship between satisfaction and turnover intention. Satisfaction had a strong negative relationship with turnover intention, while well-being remained low. The study by Nae and Choi [23] evidenced the direct relationship between job satisfaction and employee turnover. However, this also pointed out that employee well-being moderates the indirect relationship between job satisfaction and turnover. However, this moderator was significant only for a few specified occasions, such as employees having a highly secure attachment, and low counter-dependent and over-dependent attachment styles.

As per the literature, job satisfaction is an important factor in determining the impact on employee turnover. Accordingly, hypothesis one has been developed.

## Work-life balance

Work-life balance can be identified as the satisfactory co-existing of an employee's work-life and personal life. On one hand. this led to a positive influence on both employees and the organisation. On the other hand, negative work-life-balance has harmful effects on employees. Most employees had abuse alcohol due to this issue in the hospitality industry, which indirectly influences the organisation's productivity. Additionally, most women have suffered from depression due to poor work-life balance in the hospitality industry. Besides, burnout, exhaustion, and stress are common among employees with poor work-life balance. Therefore, the employee's commitment heavily depends on work-life balance, an essential requirement for employee retention [24]. This study states that it can be developed by adding more independent variables such as commitment and job satisfaction.

The highly negative work-life interference has amplified the turnover intentions of IT employees in Pakistan. They also found that the organisation that invested heavily in creating proper work-life balance recorded the lowest turnover among other organisations in the IT industry in Pakistan. Oosthuizen, Coetzee [6] revealed that the overall work-life balance had no clear influence on the satisfaction of an employee's current job. Gender was a primary separation point of work-life balance variation among employees. Female employees looked more satisfied with their work-life balance than male employees [6]. In this light, work-life balance is one part of quality work life other than career opportunities and job characteristics. Organisational embeddedness has a positive and strong relationship with work-life balance. Positive work-life balance has a negative relationship with turnover intention [25]. However, the sample of this research was based on two healthcare firms. Since the whole world is tech-driven, it is realistic to focus on the IT industry too for generalisability of findings.

According to this study, superiors' influence on work-life balance highly impacts job satisfaction. Supportiveness and the supervisor's flexibility on subordinates' help achieve the desired work-life balance for employees. As noted before, the employee turnover intention is heavily dependent on work-life balance. As such, a study on work-life balance can predict the turnover intention of an employee accurately compared to other factors. Work-life balance can be measured and categorised into three. Interference of work on personal life, work and family conflict and facilitation of work and family are those categories that the researcher suggested. The sample for the study of Kaur and Randhawa [16] was Indian private school teachers. The researcher suggested that formulating teacher-friendly policies to enhance work-life balance will reduce teachers' turnover intentions. The researcher also suggested that the imbalance workload on employees supports increasing employee turnover intentions. However, most of the employees in this study were females.

Organisations that focused on employees' proper work-life balance have recorded better efficiency, innovation, and talent retention [26]. Employee engagement and life satisfaction have been significantly mediated by the work-life balance of restaurant employees in Nevada, USA [27]. However, there are not sufficient recent researchers in Sri Lanka on work-life balance and employee retention. Therefore, taking up this study as an opportunity to research is essential. According to the above literature, hypothesis two has been constructed; work-life balance has a negative impact on employee turnover.

## Happiness

Employee happiness is a psychological feeling they have with the workplace. This is an essential factor in maintaining a successful and profitable organisation. Wright and Cropanzano [17] described happiness as phycological well-being. Personal well-being is one better way to explain employee retention. By moderating this factor, firms can achieve better employee turnover.

The workplace must be a source of happiness for employees. Unhappy employees in a workplace tend to increase employee turnover, absenteeism, low productivity, and time wasted deadlines. Creating happiness within the workplace is not a simple process. It is a comprehensive and continuous process. Happy employees generally have a fair idea of the organisation's vision, mission and values. Employees in each department should have a clear idea about their goals [28]. However, happy employees are not always productive. But they can guide and explore things without organisations forcing them. Those employees required proper career management and support to be productive.

The workplace's physical environment plays a major role in employee happiness and cheerfulness and friendliness of the physical environment are fundamentals. Employee's attitude

also has a more significant effect on happiness. Gratitude, appreciation, servant leadership from the organisation, hope and interpersonal connection are the main factors that affect the employee's positive attitude. Humour, fun and games also play a major role in keeping employees happy. Other than those factors, wellness activities, celebrations and compensation are the minor factors affecting employee happiness. Based on the above cited literature, hypothesis three can be developed; employee happiness has a negative impact on employee turnover.

## Management support

Management support is a must in the move from a good to a great company. Management stands by employees and supports them mentally and physically. Van den Heuvel, Freese [29] conducted research from the data of 699 employees at three divisions within the Dutch subsidiary of a multinational organisation. Management increased employee autonomy by supporting them to work from anywhere at any hour. This positively affected employee engagement and was negatively related to employee retention. Trust in management is a critical factor in employee turnover.

A cross-sectional survey has been conducted for front-line healthcare staff in China by Li, Mohamed [30] to measure the impact of organisational support on employee turnover intention. This study's results could verify that organisational support negatively affected employee turnover intention. Saoula and Johari [31] studied this area and determined a negative relationship between organisational support and employee turnover intention. As both of the above explained research have been conducted in non-Western countries, the findings help to complete the theoretical framework for the current study in the Sri Lankan context.

Wong and Wong [5] researched the world's most populous county, China, to identify the relationship between perceived organisational support and employee turnover. The findings suggested that trust, job security and distributive justice negatively impact employee turnover. However, China is an Asian country, and these similarities may apply to specific research findings in the Sri Lankan context.

Employee perception of management support for employee health is a factor in employee retention. Xiu, Dauner [32] studied this area with employees' data from a public university, which was the first empirical examination of organisational support for employee health and retention. This kind of approach leads to building trust with employees. Moreover, these findings are essential to human resource managers who are willing to promote employee well-being at the workplace. Hypothesis four has been developed based on above discussed literature.

## Career management

Initiatives must carry out different strategies for old and young employees because their priorities are different. Digest [18] discloses that young employees are impressed by flexible working opportunities, career advancement, positive working relationships and inclusive management forms. Young employees are more likely to be talented, leading to an organisation's success and they can also become key workers in the company.

Saoula and Johari [31] researched the effect of personality traits (big five) on employee turnover intention. The researchers state that the relationship between the big five personality traits and turnover intention will support early prediction of employee turnover intentions. Identifying employee's personalities and helping them to find the most suitable job role is a long-term process, though it will be highly advantageous for both employees and the organisation.

Rawashdeh and Tamimi [33] focused on the latest management developments of leading organisations worldwide. They state that there is a strong relationship between the availability

of training and supervisor support for training and organisational commitment. Further, they proved that there is a strong negative association between organisational commitment and employee retention. These research findings verify the social exchange theory [34]. However, the research suggested that the above study can improve by adding new factors like motivation and co-worker support for training. Hypothesis five has been developed by concluding the above explained literature.

### Innovative work behaviour

Innovative behaviour is a leading factor in gaining a competitive advantage. Shih, Posthuma [35] investigated the negative impacts of innovative work behaviour on employee turnover and conflict with co-workers. According to the studies, there is a positive relationship between innovative work behaviour and employee turnover. Further, it found that perceived distributive fairness can negatively moderate this relationship. However, the writer has suggested extending the research to different geographical locations and industries.

The organisational learning culture is a key factor for innovative work behaviour. Saoula, Fareed [36] conducted research in Malaysia, a developing country in Asia to examine the relationship between organisational learning culture and employee turnover intention. The organisational learning culture improves learning capability, supports sustainable development, and affects organisation's positive changes. As organisational learning culture and employee turnover intention have a negative relationship, the result helps to identify the impact of innovative work behaviour. According to the existing literature, limited studies have been conducted on this topic.

Agarwal, Datta [4] conducted research with managerial employees in India to examine the relationship between innovative work behaviour and employee turnover. This study asserted that the variables have an inverse relationship. As innovative work behaviour examinations in an Asian county country like India, it is important to consider this variable in this model. With the presence of the above mentioned literature, hypothesis six has been formulated.

### Leader member exchange

As per many leadership methods, leader member exchange depends on the leadership style. Tobias M. Huning [37] conducted research to identify the effect of servant leadership on employee turnover. Servant leadership supports employee empowerment, development, interpersonal acceptance, and courage. This study found that servant leadership negatively impacts employee turnover. However, this leadership style does not directly affect employee retention. Gyensare, Kumedzro [38] studied the impact of transformational leadership on employee turnover. This type of leadership supports work engagement of the employee, and it negatively relates to employee retention. Considering both aspects, the study found that increasing work engagement is vital to curtail employee retention.

Leader support is an indirect factor in employee retention. According to the studies, employee engagement and work-life balance act as mediation for perceived supervisor support and employee turnover relationship [16]. The supervisor supports the career success of employees and it affects both directly and indirectly the career success of the employee and retention one year later [9]. Therefore, this study shows that co-worker support has a significantly positive impact on employee turnover. However, these results maintained the diversity of the sample. As this has been examined in India, a South Asian country, the same results can apply to the Sri Lankan context. Based on the above-mentioned literature, hypothesis seven has been developed; leader member exchange has a negative impact on employee turnover.

## Co-worker support

Co-worker support will be in both formal and informal ways and in two different forms, emotional support, and instrumental support. The support of co-workers enhances the confidence level of the employee. Further, it helps to accept challenges in the work environment. Kmieciak [19] has worked on research to identify the effect of co-worker support on employee retention in the IT industry. However, a significantly negative impact was not evident on co-worker support. As this is a recently published research paper, the results are more valuable to the current research. The researcher has investigated more about the impact of subordinates' support. Here, the analysis has been done only with 118 employees from a Polish software company. Considering the above limitations enables researchers to further study this topic with a larger sample size for generalisability of findings.

Abugre and Acquaah [39] researched in Ghana to identify the relationship between co-worker relationships and employee retention. The findings of this research imply that co-worker support is negatively associated with employee turnover. It further stated that cynicism of the employee is positively associated with employee turnover. The speciality of this research is identifying the importance of encouraging co-worker support rather than employee cynicism. These newly published research results can be used along with all other variables that affect employee turnover. According to the above literature, hypothesis eight has been constructed.

These studies have a common limitation in gathering more independent variables and analysing the impact. Therefore, a need exists to measure the effect of job satisfaction, work-life balance, happiness, management support, career management, innovative work behaviour, leader member exchange, and co-worker support together on employee turnover.

In Sri Lanka, no research has so far considered all eight factors affecting employee turnover in one study. With the above-mentioned literature findings, this study assists the government in identifying the impact of every factor on employee turnover in startups in Sri Lanka.

## Data and methodology

### Data

This study was reviewed and approved by Sri Lanka Institute of Information Technology Business School and the Sri Lanka Institute of Information Technology ethical review board. Data were collected through a questionnaire using both online and manual channels. Each individual in this study gave verbal consent prior to the formal interview. The data was collected from August to September 2022 (S1 Appendix). The authors directly distributed the questionnaire. Moreover, authors could contact management in startups and distribute the questionnaire in their organisation. The questionnaire is composed of ten (10) sections. The first part of the questionnaire was designed to collect the demographic characteristics of the correspondents. The second to ninth sections focused on independent variables, job satisfaction, work-life balance, happiness, management support, career management, innovative work behaviour, leader member exchange, and co-worker support. Finally, the tenth section was designed to identify employee turnover indicator. A minimum of four questions was added under each indicator. The researchers facilitated anonymously answering all the questions in the questionnaire. The participants should be a part of startup and he/she should consider the behaviour and culture of that startup when answering the questions. All nine indicators were covered by Likert scale questions from 1 to 5 rating scale, depicting (1) strongly disagree to (5) strongly agree to collect respondents' attitudes and opinions. Each respondent took about 10–15 minutes to complete answering the questionnaire and took approximately 5–7 minutes to fill out the questionnaire. Furthermore, the average values were calculated to measure the value given by respondents for each indicator. The data file used for the study is presented in S2 Appendix.

PricewaterhouseCoopers [15] statistics determined the study's population and it explained the total number of elements to be focused on in this study. The researchers applied a random sampling method, mainly employees who are a part of or have been a part of the startup. This sampling technique was appropriate because it was free of bias. The sample size was selected by referencing the Krejcie and Morgan sampling table and Calculator.net [40] with a confidence level of 95% and 7% of margin of error. The calculation results indicated a minimum of 171 professionals. A stepwise ordered probit analysis method was used as the selected variables are widely used indicators for employee turnover; therefore, a micro-level analysis was required to study how these variables impact. A pilot survey was conducted to identify whether the purpose of the questions was clear to the respondents.

The data used for the estimation include 83 low employee turnover, 79 moderate employee turnover and 68 high employee turnovers of employees in Sri Lankan startups. The initial estimation results are presented in Table 1.

The mean values of all independent variables are greater than 2.5. Respondents were further grouped as per demographic and geographic characteristics. The respondents' gender identity ratio is nearly 1: 2. When considering the age groups, most are in 20–30 years. Many employees in startup companies are in their twenties and are graduates. The respondents represent all the districts in Sri Lanka, most of which are from Kalutara, Colombo, Galle and Matara districts.

## Research framework and hypothesis

The conceptual framework was developed with the literature review and existing knowledge, as illustrated in Fig 2. This model was developed with the combination of eight hypotheses. These independent variables have been identified as critical factors that impact employee turnover.

The following hypotheses have been developed in line with the research framework.

**Hypothesis 1**: Job satisfaction has a negative impact on employee turnover in startups in Sri Lanka.

**Hypothesis 2**: Work-life balance has a negative impact on employee turnover in startups in Sri Lanka.

**Hypothesis 3**: Happiness has a negative impact on employee turnover in startups in Sri Lanka.

**Hypothesis 4**: Management support has a negative impact on employee turnover in startups in Sri Lanka.

**Hypothesis 5**: Career management has a negative impact on employee turnover in startups in Sri Lanka.

**Hypothesis 6**: Innovative work behaviour has an impact on employee turnover in startups in Sri Lanka.

**Hypothesis 7**: Leader member exchange has an impact on employee turnover in startups in Sri Lanka.

**Hypothesis 8**: Co-worker support has a negative impact on employee turnover in startups in Sri Lanka.

## Methodology

This study focuses on the demographical variables that affect employee turnover. For this, the present study's authors considered employee feedback concerning Sri Lankan startups. The

**Table 1. Characteristics of employee turnover in Sri Lankan startups.**

| Variable | Analytics sample (N = 230) | |
|---|---|---|
| | **(Means if numerical)** | **Standard deviations** |
| **Dependent variable—Employee Turnover (ET)** | | |
| Low | 36.09% | |
| Moderate | 34.35% | |
| High | 29.57% | |
| **Independent variables** | | |
| Job Satisfaction (JS) | 3.8343 | 0.0639 |
| Work-life Balance (WLB) | 4.0583 | 0.0642 |
| Happiness (H) | 4.0017 | 0.0637 |
| Management Support (MS) | 4.0235 | 0.0730 |
| Career Management (CM) | 3.8989 | 0.0694 |
| Innovative Work Behaviour (IWB) | 3.8565 | 0.0704 |
| Leader Member Exchange (LMX) | 4.0924 | 0.0711 |
| Co-Worker support (CWS) | 4.0913 | 0.0689 |
| **Gender identity** | | |
| Male | 68.26% | |
| Female | 31.74% | |
| **Age group (in years)** | | |
| 20–30 | 88.70% | |
| 31–40 | 7.39% | |
| 41–50 | 2.61% | |
| Above 50 | 1.30% | |
| **Educational/Professional qualifications** | | |
| Passed G.C.E. O/L or G.C.E. A/L or equivalent | 3.91% | |
| Passed certificate or diploma level | 20.43% | |
| Passed degree | 58.26% | |
| Passed postgraduate | 17.39% | |
| **Work status** | | |
| Full time | 93.04% | |
| Part time | 6.96% | |
| **Geographical locations (Districts)** | | |
| Ampara | 3.48% | |
| Badulla | 0.87% | |
| Colombo | 24.78% | |
| Galle | 12.17% | |
| Gampaha | 6.52% | |
| Hambantota | 0.87% | |
| Jaffna | 0.43% | |
| Kalutara | 27.83% | |
| Kandy | 1.30% | |
| Kegalle | 0.43% | |
| Kurunegala | 2.17% | |
| Matale | 0.87% | |
| Matara | 10.00% | |
| Nuwara Eliya | 2.61% | |
| Polonnaruwa | 0.43% | |
| Puttalam | 1.30% | |

(*Continued*)

**Table 1.** (Continued)

| Variable | Analytics sample (N = 230) | |
|---|---|---|
| | (Means if numerical) | Standard deviations |
| **Dependent variable—Employee Turnover (ET)** | | |
| Rathnapura | 0.87% | |
| Ratnapura | 2.17% | |
| Trincomalee | 0.87% | |

Source: Authors' compilation based on survey data.

ordered probit regression determines the significant variables [41]. The probit model is an estimation technique for equations with dummy dependent variables that avoids the unboundedness problem of the linear probability model by using a variant of the cumulative normal distribution [42]. Further, this study examines the likelihood of three types of employee turnover. Accordingly, employee turnover is divided into three categories, considering the equality of data for each category based on employee turnover.

Group 1 (y = 1): low = mean value of the employee turnover less than 1.50

Group 2 (y = 2): moderate = mean value of the employee turnover greater than 1.5 and less than or equal to 2.25

Group 3 (y = 3): high = mean value of the employee turnover greater than 2.25 and less than or equal to 5

The following equation represents the general form of the ordered probit model.

$$y_i^* = x_i'\beta + \varepsilon_i \tag{1}$$

The $y_i$ value represents i$^{th}$ value of the dependent variable, employee turnover and $x_i$ represents the i$^{th}$ common independent variable. The $\beta$ value is a vector parameter and $\varepsilon_i$ considered as the normally distributed random error term with a zero mean. The following ordered

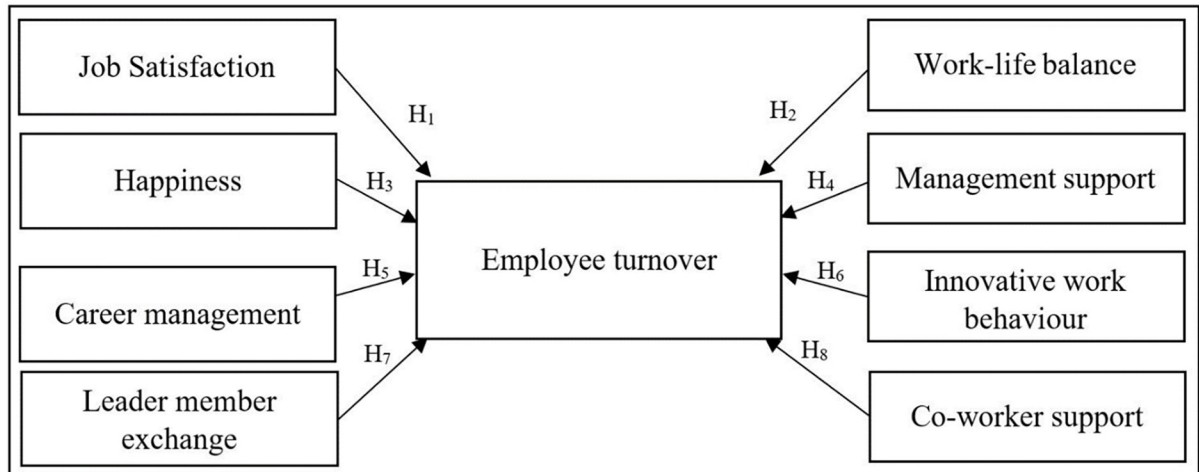

**Fig 2. Conceptual framework.** Source: Authors' compilation.

probit model has been developed by detailing the general equation.

$$
\begin{aligned}
Y(ET = 1, 2, 3) \\
= X_i(\beta_0 + \beta_1 \ ln(JS) + \beta_2 \ ln(WLB) + \beta_3 \ ln(H) + \beta_4 \ ln(MS) + \beta_5 \ ln(CM) \\
+ \beta_6 \ ln(IWB) + \beta_7 \ ln(LMX) + \beta_8 \ ln(CWS)) + \varepsilon_i
\end{aligned} \quad (2)
$$

Table 2 indicates the variables explained in previous literature and definitions of the previously mentioned equation that affects employee retention. The forward stepwise regression model has been used to analyse the data set.

## Results and discussions

It is mandatory to test the internal consistency reliability before data analysis. The most common measure of reliability is Cronbach's alpha (α) value, which determines whether the internal instruments are constant [43]. The reliability results for each indicator are presented in Table 3. As all the Cronbach alpha values are greater than 0.6 scale reliability coefficients, all variables in this study are acceptable.

In the first step, the initial ordered probit model was executed, and this model explained 73% of the variation in employee retention by the variation in independent variables. S3 Appendix contains the table of the initial ordered probit regression model. The ordered probit model forwarded with the forward stepwise technique to identify the exact number of variables that impact employee turnover. A forward stepwise technique was adopted for the variable selection in each specification. Here, the new variables for selection were considered with a p-value < 0.20 and the previously selected variable for removal with a p-value ≥ 0.25. Three different model diagnostic criteria were considered in assessing the reliability of the results. The forward stepwise methodology suggested that the significance of the existing variables could be increased by adding more variables to the model. Marginal effects were separately calculated for low, moderate, and high employee turnover. Table 4 presents the final estimation

**Table 2. Variable definitions.**

| Variable | Description | Expected sign (s) |
|---|---|---|
| ET | Dummy variable to capture the turnover of employees in startups where low is denoted by 1, moderate as 2 and high as 3. | (-) |
| JS | Job satisfaction. Five-point Likert scale variable with extremes "rarely– 1" to "always– 5" will be used. | (-) |
| WLB | Five-point Likert scale variable with extremes "rarely– 1" to "always– 5" to measure work-life balance of employees. | (-) |
| H | Employee's happiness. Five-point Likert scale variable with extremes "rarely– 1" to "always– 5" will be used. | (-) |
| MS | Five-point Likert scale variable with extremes "rarely– 1" to "always– 5" to measure management support to the employee. | (-) |
| CM | Career management. Five-point Likert scale variable with extremes "rarely– 1" to "always– 5" will be used. | (-) |
| IWB | Five-point Likert scale variable with extremes "rarely– 1" to "always– 5" to measure innovative work behaviour of the employee. | (+/-) |
| LMX | Leader member exchange. Five-point Likert scale variable with extremes "rarely– 1" to "always– 5" will be used. | (+/-) |
| CWS | Five-point Likert scale variable with extremes "rarely– 1" to "always– 5" to measure employee's co-worker support. | (-) |

Source: Authors' compilation.

**Table 3. Internal consistency.**

| Item | Number of items | Average interitem covariance | Scale reliability coefficient (Cronbach alpha for dimensions) |
|------|-----------------|------------------------------|--------------------------------------------------------------|
| *ET* | 9 | 0.7698 | 0.9647 |
| *JS* | 2 | 0.7951 | 0.9145 |
| *WLB* | 2 | 0.7444 | 0.8835 |
| *H* | 2 | 0.9076 | 0.9134 |
| *MS* | 2 | 0.9840 | 0.9148 |
| *CM* | 2 | 0.9740 | 0.9284 |
| *IWB* | 2 | 0.8397 | 0.8433 |
| *LMX* | 2 | 0.9228 | 0.8997 |
| *CWS* | 2 | 0.7654 | 0.8593 |

Analytical sample (N = 230)

Source: Authors' calculation based on survey data

results of the ordered probit model and illustrates the substantive effects of the independent variables. Here, 71.74% of the variation in employee turnover is explained by the variation in job satisfaction, LMX and co-worker support, considering the sample size and independent variables.

Looking at the signs of the marginal effects in Table 4, overall, high employee turnover is negatively associated with job satisfaction, co-worker support, and innovative work behaviour, whereas high employee turnover is positively associated with leader member exchange.

To control for the potential effect on different levels of employee turnover, the age factor was also included in the model, the coefficient of which implies that high employee turnover is

**Table 4. Final ordered probit regression results.**

| Variable | Estimate | Robust SE | Marginal effects (in percentages) | | |
|----------|----------|-----------|-----------------|-----------------|-----------------|
| | | | Low ET ($Y = 1$) | Moderate ET ($Y = 2$) | High ET ($Y = 3$) |
| ln*JS* | -1.3829** | 0.6500 | 0.4716** | -0.0387 | -0.4329** |
| ln*IWB* | -0.8552* | 0.4856 | 0.2917 | 0-.0239 | -0.2677* |
| ln*LMX* | 1.3808*** | 0.4127 | -0.4709*** | 0.0386 | 0.4323*** |
| ln*CWS* | -1.1872*** | 0.4264 | 0.4049*** | -0.0332 | -0.3717*** |
| ln*CM* | -0.6897 | 0. 4915 | 0.2352 | -0.0193 | -0.2159 |
| *Socio-demographic characteristics* | | | | | |
| G_Male | 0.4440*** | 0.1723 | -0.1569** | 0.0264 | 0.1305*** |
| A_20_30 | 0.8527 | 0.5439 | -0.3220 | 0.1209 | 0.2011** |
| A_31_40 | 1.6657*** | 0.6027 | -0.3133*** | -0.2817** | 0.5950*** |
| **Ancillary parameters** | | | Marginal effects after ordered probit | | |
| $\hat{\gamma}_1$ | -2.9373 | 0.6738 | 0.2877 | 0.4691 | 0.2431 |
| $\hat{\gamma}2$ | -1.6810 | 0.6571 | | | |
| Pseudo R$^2$ | 0.7174 | | | | |
| Log likelihood | -197.1130 | | | | |
| Number of observations | 230 | | | | |

Note

*** significant at the 1% level

** significant at the 5% level and * significant at the 10% level.

Source: Authors' calculation based on surveying data.

0.20 points and 0.60 points for the 20–30 years age range and 31–40 years age range, respectively. Employee turnover in 31–40 years age range employees is higher than that of other age ranges.

The marginal effects of the psychographic variables reveal that a 1% increase in job satisfaction increases the probability of low employee turnover by 0.47 percentage points. Similarly, 1% increase in job satisfaction decreases the probability for high employee turnover by 0.43 percentage points. With this observation, it can be stated that improving job satisfaction will highly affect to reduce high employee turnover. These results verify the existing statements indicating that job satisfaction has the highest significant and negative estimate value.

The estimated marginal effect of low employee turnover is 0.47 percentage points higher for employees in Sri Lankan Startups with a 1% increase in leader member exchange. High employee turnover is associated with leader member exchange increasing probability by 0.43. However, this study reflects similar findings to those of Tymon, Stumpf [9]. The reason behind the positive relation is employees learn fast and get qualified with the support of their leaders and then quit the company within the next few years.

Both leader member exchange and co-worker support are significant at the 99% level of employee turnover in the Sri Lankan context. When considering the independent variables for employee turnover in startups in Sri Lanka, co-worker support is a critical factor in determining the level of employee turnover. The 1% increase in co-worker support will also increase the probability of low employee turnover by 0.40 percentage points. But concurrently, change in co-worker support will negatively impact high employee turnover. The results ensure that encouraging co-worker support is crucial rather than employee cynicism.

Innovative work behaviour is one of the most critical factors in employee turnover. With a 1% increase in innovative work behaviour, the estimated marginal effect of high employee turnover is 0.27 percentage points lower for employees in Sri Lankan startups. The results of Shih, Posthuma [35] indicate a positive relationship exists between innovative work behaviour and employee turnover. However, this study concludes by emphasising the importance of retaining the innovative employees to remain competitive in the industry. For this, startups need to improve and enhance employees' innovative behaviour and, concurrently, to prevent such employee retention.

Entrepreneurs are the founders of startups. Employees' entrepreneurial dreams positively affect employee intention to startups. Employees in the startups also will have an ideation to start their own business. According to the study by Li, Li [44] the mediating role of employees' entrepreneurial self-efficacy and the moderating role of job embeddedness in the influence of entrepreneurial dreams on employees' turnover intention to startup.

## Conclusion

The main objective of this research is to analyse the impact of critical and newly identified factors on employee turnover in one study. This issue occurs when employees leave the company by giving short notice or quitting unexpectedly. The analysis found that gender and age impact employee turnover in startups in Sri Lanka. In startups, many employees are in the 20 to 30 years age range. Employees between 31 and 40 years show a higher tendency to leave the startups. In Sri Lanka, only 8% of startups have been in operation for more than five years [15], indicating that the businesses are not stabilised and are still in its early stages. To prevent employee turnover, startups must improve employee job satisfaction. As per the findings, increasing job satisfaction has a significant impact on reducing employee turnover. For most employees in startups, it is their first job. During this time, employees gain work experience and become experts in the field. The leaders allocate much time to train their human resources

and the company should gain strategic benefits from this investment. The results of the study prove that leader member exchange has a positive impact on employee turnover, as verified by Tymon, Stumpf [9] too about this relationship. To overcome this situation, as managers, it is vital to discuss with employees about their career paths, employee interests and company's business plans while improving their technical skills and experience. This way, the mutual interest of both the employee and the company can be identified and handled. It also builds trust between the company and the employees. Regular support environment and ease of doing business is 66% highly important factor for the success of Sri Lankan startups [15]. This environment can be easily created with the level of co-worker support to the employee. Employee turnover can be more costly than a startup can imagine, with disruptions to business operations when their employees' suddenly quit jobs. Therefore, it is must to attain above discussed facts. These results and discussions can be taken as insights to better understand and curtail employee turnover. This study will assist Sri Lankan startups where their skilled employees, who are also experts plausibly remain, enabling the businesses to expand to new markets. Usually, issues relevant to profit-making and business performance, such as a drop in sales and manufacturing are identified by startups. However, employee turnover is generally not identified as an organisational issue.

## Theoretical implications

The current study empirically investigated the impact of job satisfaction, innovative work behaviour, co-worker support and leader member exchange on employee turnover. According to the authors' knowledge, no prior studies were conducted considering the combined impact of all the independent variables on employee turnover. Therefore, this study strengthens the literature by demonstrating how job satisfaction, innovative work behaviour, co-worker support and leader member exchange impact employee turnover in Sri Lankan startups.

The findings reveal that job satisfaction has a negative impact on employee turnover. This finding is consistent with the previous study, job satisfaction significantly predicted employee turnover [6]. This study consolidates past findings that male employees have higher turnover intention than female employees. Female employees have comparatively higher-level job satisfaction [8]. This study implies that employees age 31 to 40 years have high employee turnover intention. The research findings are similar to Lu, Lu [8]; the older employees have high intentions to leave the company.

## Practical implications

The study's findings illustrate the importance of job satisfaction, innovative work behaviour, co-worker support and leader member exchange in affecting employee turnover in startups. This study provides managerial insights on lowering employee turnover in Sri Lankan startups. First, startups need to be aware that experienced employees in startups can be easily taken by well-established companies because, later, they have hand on experience and skills. Therefore, it is important to implement strategies for a solid career development plan, career growth, personal status, and employee recognition. As job satisfaction can predict employee turnover, it is a must to measure those indicators and maintain a favourable level at all times.

Innovative work behaviour is increasingly becoming a significant factor in employee retention. As good startups are a blend of creativity and competitive advantage, it is a must to focus on the *IWB* of the employee. *LMX* is a turning point for expanding the business. More importantly, healthy *LMX* can boost employees' work engagement. This healthy level can maintain by conducting regular meetings, training programs and informal mentorship with employees' immediate supervisors [8]. Further, management can allow employees at all levels to present

their fresh ideas and incorporate them to influence organisation's decision making process. These processes can lower employee hierarchy and build strong relationships while recognising them in the company.

It is important to retain trained and skilled employees who started their career paths in the organisation. Such employees can drive the organisation to success. While measuring employees' job satisfaction, managers nee to conduct standard ways on performance and improvements of the organisation. It is better if companies can create their key performance indicators because it will help protect the organisation's core values while expanding the company. Furthermore, having a flexible approach to work in an organisation culture will increase the trust between employees and the organisation. Giving the freedom to take risks and not allowing them to feel alone during work will give value to employees. Finally, all the above actions will strongly impact reducing employee intention to leave the organisation.

### Research limitations and future research directions

Further research can improve the study as follows. First, this research includes feedback from 230 employees. More than one-third of these employees are from the IT industry. Since Sri Lankan startups are technology-driven, this ratio is more reliable. However, this research can be generalised by obtaining employees' feedback from other industries. Secondly, in this questionnaire, the minimum number of questions for independent factors is four. This is to minimise the possibility of demotivating the employee by giving a lengthy and complex questionnaire. Therefore, in future, researchers can design questionnaires incorporating more questions to cover a wider range of independent factors, including open-ended ones. Thirdly, in this sample, many employees were in their twenties, and most hadn't worked for more than two companies (i.e. employers). As such, it is reasonable to assume that participants' response is somewhat limited to obtain the broader picture of the research problem. Future researchers can focus on different age groups and analyse the same factors concerning employee retention. Finally, new research can be executed by adopting a case study approach (including case studies representing various types of industries etc), such as employees in multinational companies.

### Supporting information

**S1 Appendix. Questioner.**
(DOCX)

**S2 Appendix. Data file.**
(XLSX)

**S3 Appendix. Initial ordered probit regression results.**
(DOCX)

**S4 Appendix. Literature summary.**
(DOCX)

### Acknowledgments

The authors would like to thank Ms. Gayendri Karunarathne for proof-reading and editing this manuscript.

### Author Contributions

**Conceptualization:** Lakshmi Kanchana, Ruwan Jayathilaka.

**Data curation:** Lakshmi Kanchana.

**Formal analysis:** Lakshmi Kanchana.

**Methodology:** Lakshmi Kanchana, Ruwan Jayathilaka.

**Software:** Lakshmi Kanchana.

**Supervision:** Ruwan Jayathilaka.

**Validation:** Lakshmi Kanchana, Ruwan Jayathilaka.

**Visualization:** Lakshmi Kanchana.

**Writing – original draft:** Lakshmi Kanchana, Ruwan Jayathilaka.

**Writing – review & editing:** Ruwan Jayathilaka.

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
