## [Decision Letter · Decision Letter 0]

21 Dec 2022

PONE-D-22-30684How are employee turnover intentions created in Sri Lankan Startups?PLOS ONE

Dear Dr. Ruwan Jayathilaka,

Thank you for submitting your manuscript to PLOS ONE. After careful consideration, we feel that it has merit but does not fully meet PLOS ONE’s publication criteria as it currently stands. Therefore, we invite you to submit a revised version of the manuscript that addresses the points raised during the review process.

Reviewer#1Abstract: Rewrite the abstract after manuscript correction and provide picture of whole study.

Introduction

In the first paragraph of introduction used only this (2016) citation. This citation not justify the paragraph.

Introduction paragraph is not justifying the problem and bag-round of study. Revised the introduction and use the recent citations to justify and logically make connection with them.

 In the introduction (second paragraph) , the contribution of study is confused with variable relationships; why are these relationships a contribution of study? Need strong justification.

 Overall, I suggest a major rewrite of the introduction. It should provide an overview of and focus on one issue with recent citations.

Literature review

Revised all literature variables and link with variables with new citation

In the literature, justify these hypotheses with literary support.

In literature, justify the conceptual model and theoretical gap.

Methodology

Where is the total population? How did you choose the sample size? And how did you choose which method, unit of analysis, and research technique to use? Provide justification. Why is this method appropriate for this data set?

General: identifying flaws in the study's design (revised methodology section) and justifying technique

Discussion

Write the theoretical contribution related to a model. Reviewer#2

Abstract:

Mention the scope of the study, the population, simple size, data collected from…..

Mention the analysis technique/ tool used in the study

Introduction:

The introduction is not clear and very less literature is used. Follow these instruction: The introduction should briefly place the study in a broad context and highlight why it is important. It should define the purpose of the work and its significance.

The current state of the research field should be reviewed carefully and key publications cited. Briefly mention the main aim of the work and highlight the main conclusions. Keep the introduction comprehensible to scientists working outside the topic of the paper.

What is the main research focus?? Firm performance? Employee retention? Employee turnover intention?

Focus more on the main issue of the study

Make the theoretical and practical gaps more clear ?

Why Sri Lanka?

Why stratup s in Sri Lanka? Why employee working in startups in Sri Lanka

What was the key motivation behind focusing on factors affecting employee turnover intention in stratups in Srilanka?

Please, properly justify why the selected variables are included in the model. How did you derive the 08 variables ??

As ,many studies conducted in the world and in Sri Lanka about this topic, what us the main contribution of your study?

The paper should incorporate a more solid argumentation that allows to justify the reason that allows to select the explanatory variables that are considered in the empirical analysis.

Literature and hypotheses development"

Improve the argumentation of hypothesis. Whether, the hypotheses are formulated separately or after the literature review of each section, it should be properly argued.

Each hypothesis should be formulated at the end of a literature section of the each variable presenting the different findings that have been made throughout the literature. With these arguments a reasoning should be developed in a certain direction and the conclusion of that reasoning should be the formulated hypothesis. In the current version of this manuscript the authors are including different aspects of previous literature, but it does not exist any convincing storyline in any direction.

Highlight controversial and diverging hypotheses when necessary.

Researcher should include a summary table / review on studies conducted on Employee Turnover Intention in Sri Lanka to support the literature and arguments.

Below papers has some interesting implications and understanding of concepts and relations that you could discuss in your introduction and literature review and how it relates to your work:

-Li, M., Li, J., Chen, X. “Employees’ Entrepreneurial Dreams and Turnover Intention to Start-Up: The Moderating Role of Job Embeddedness”, 2022, International Journal of Environmental Research and Public Health 19(15),9360

- Saoula, O.,Johari, H, “The mediating effect of organizational citizenship behavior on the relationship between perceived organizational support and turnover intention: A proposed framework” International Review of Management and Marketing, 2016, 6(7), pp. 345–354

- Saoula, O., Johari, H., Bhatti, M.A “The mediating effect of organizational citizenship behaviour on the relationship between personality traits (Big Five) and turnover intention: A proposed framework”, International Business Management, 2016, 10(20), pp. 4755–4766.

Zito, M., Emanuel, F., Molino, M., Cortese, C. G., Ghislieri, C., & Colombo, L. (2018). Turnover intentions in a call center: The role of emotional dissonance, job resources, and job satisfaction. PloS one, 13(2), e0192126.

- Saoula, O., Johari, H., Fareed, M, “A conceptualization of the role of organisational learning culture and organisational citizenship behaviour in reducing turnover intention”, Journal of Business and Retail Management Research, 2018, 12(4), pp. 126–133

- Saoula, O., Fareed, M., Ismail, S.A., Husin, N.S., Hamid, R.A, “A conceptualization of the effect of organisational justice on turnover intention: The mediating role of organisational citizenship behaviour”, International Journal of Financial Research, 2019, 10(5), pp. 327–337.

Poku, C. A., Alem, J. N., Poku, R. O., Osei, S. A., Amoah, E. O., & Ofei, A. M. A. (2022). Quality of work-life and turnover intentions among the Ghanaian nursing workforce: A multicentre study. PloS one, 17(9), e0272597.

- Saoula, O., Fareed, M., Hamid, R.A., Al-Rejal, H.M.E.A., Ismail, S.A, “The moderating role of job embeddedness on the effect of organisational justice and organisational learning culture on turnover intention: A conceptual review”, Humanities and Social Sciences Reviews, 2019, 7(2), pp. 563–571

-Li, Q., Mohamed, R., Mahomed, A., & Khan, H. (2022). The Effect of Perceived Organizational Support and Employee Care on Turnover Intention and Work Engagement: A Mediated Moderation Model Using Age in the Post Pandemic Period. Sustainability, 14(15), 9125.

- Amin, M., Othman, S.Z., Saoula, O, “The Effect of Organizational Justice and Job Embeddedness on Turnover Intention in Textile Sector of Pakistan: The Mediating Role of Work Engagement” Central Asia and the Caucasus, 2021, 22(5), pp. 930–950

Methodology:

How experiment was conducted?

How participants were recruited?

What are the instructions of experiment?

How much was time given to each participant?

Others:

What are the theoretical implications of the study ?

Practical implications needs further discussion.

Add/ involve more recent citations/studies where necessary

We look forward to receiving your revised manuscript.

Kind regards,

Muhammad Fareed, Ph.D

Academic Editor

PLOS ONE

Journal Requirements:

2. In the ethics statement in the Methods, you have specified that verbal consent was obtained. Please provide additional details regarding how this consent was documented and witnessed, and state whether this was approved by the IRB.

Additional Editor Comments:

The paper is generally well written and structured. However, I believe that paper has some shortcomings in terms of

Abstract: Rewrite the abstract after manuscript correction and provide picture of whole study.

Introduction

In the first paragraph of introduction used only this (2016) citation. This citation not justify the paragraph.

Introduction paragraph is not justifying the problem and bag-round of study. Revised the introduction and use the recent citations to justify and logically make connection with them.

In the introduction (second paragraph) , the contribution of study is confused with variable relationships; why are these relationships a contribution of study? Need strong justification.

Overall, I suggest a major rewrite of the introduction. It should provide an overview of and focus on one issue with recent citations.

Literature review

Revised all literature variables and link with variables with new citation

In the literature, justify these hypotheses with literary support.

In literature, justify the conceptual model and theoretical gap.

Methodology

Where is the total population? How did you choose the sample size? And how did you choose which method, unit of analysis, and research technique to use? Provide justification. Why is this method appropriate for this data set?

General: identifying flaws in the study's design (revised methodology section) and justifying technique

Discussion

Write the theoretical contribution related to a model.

Reviewers' comments:

Reviewer's Responses to Questions

**Comments to the Author**

1. Is the manuscript technically sound, and do the data support the conclusions?

Reviewer #1: Partly

Reviewer #2: Partly

2. Has the statistical analysis been performed appropriately and rigorously? 

Reviewer #1: Yes

Reviewer #2: I Don't Know

3. Have the authors made all data underlying the findings in their manuscript fully available?

Reviewer #1: Yes

Reviewer #2: No

4. Is the manuscript presented in an intelligible fashion and written in standard English?

Reviewer #1: Yes

Reviewer #2: Yes

5. Review Comments to the Author

Reviewer #1: Abstract:

Mention the scope of the study, the population, simple size, data collected from…..

Mention the analysis technique/ tool used in the study

Introduction:

The introduction is not clear and very less literature is used. Follow these instruction: The introduction should briefly place the study in a broad context and highlight why it is important. It should define the purpose of the work and its significance.

The current state of the research field should be reviewed carefully and key publications cited. Briefly mention the main aim of the work and highlight the main conclusions. Keep the introduction comprehensible to scientists working outside the topic of the paper.

What is the main research focus?? Firm performance? Employee retention? Employee turnover intention?

Focus more on the main issue of the study

Make the theoretical and practical gaps more clear ?

Why Sri Lanka?

Why stratup s in Sri Lanka? Why employee working in startups in Sri Lanka

What was the key motivation behind focusing on factors affecting employee turnover intention in stratups in Srilanka?

Please, properly justify why the selected variables are included in the model. How did you derive the 08 variables ??

As ,many studies conducted in the world and in Sri Lanka about this topic, what us the main contribution of your study?

The paper should incorporate a more solid argumentation that allows to justify the reason that allows to select the explanatory variables that are considered in the empirical analysis.

Literature and hypotheses development"

Improve the argumentation of hypothesis. Whether, the hypotheses are formulated separately or after the literature review of each section, it should be properly argued.

Each hypothesis should be formulated at the end of a literature section of the each variable presenting the different findings that have been made throughout the literature. With these arguments a reasoning should be developed in a certain direction and the conclusion of that reasoning should be the formulated hypothesis. In the current version of this manuscript the authors are including different aspects of previous literature, but it does not exist any convincing storyline in any direction.

Highlight controversial and diverging hypotheses when necessary.

Researcher should include a summary table / review on studies conducted on Employee Turnover Intention in Sri Lanka to support the literature and arguments.

Below papers has some interesting implications and understanding of concepts and relations that you could discuss in your introduction and literature review and how it relates to your work:

-Li, M., Li, J., Chen, X. “Employees’ Entrepreneurial Dreams and Turnover Intention to Start-Up: The Moderating Role of Job Embeddedness”, 2022, International Journal of Environmental Research and Public Health 19(15),9360

- Saoula, O.,Johari, H, “The mediating effect of organizational citizenship behavior on the relationship between perceived organizational support and turnover intention: A proposed framework” International Review of Management and Marketing, 2016, 6(7), pp. 345–354

- Saoula, O., Johari, H., Bhatti, M.A “The mediating effect of organizational citizenship behaviour on the relationship between personality traits (Big Five) and turnover intention: A proposed framework”, International Business Management, 2016, 10(20), pp. 4755–4766.

Zito, M., Emanuel, F., Molino, M., Cortese, C. G., Ghislieri, C., & Colombo, L. (2018). Turnover intentions in a call center: The role of emotional dissonance, job resources, and job satisfaction. PloS one, 13(2), e0192126.

- Saoula, O., Johari, H., Fareed, M, “A conceptualization of the role of organisational learning culture and organisational citizenship behaviour in reducing turnover intention”, Journal of Business and Retail Management Research, 2018, 12(4), pp. 126–133

- Saoula, O., Fareed, M., Ismail, S.A., Husin, N.S., Hamid, R.A, “A conceptualization of the effect of organisational justice on turnover intention: The mediating role of organisational citizenship behaviour”, International Journal of Financial Research, 2019, 10(5), pp. 327–337.

Poku, C. A., Alem, J. N., Poku, R. O., Osei, S. A., Amoah, E. O., & Ofei, A. M. A. (2022). Quality of work-life and turnover intentions among the Ghanaian nursing workforce: A multicentre study. PloS one, 17(9), e0272597.

- Saoula, O., Fareed, M., Hamid, R.A., Al-Rejal, H.M.E.A., Ismail, S.A, “The moderating role of job embeddedness on the effect of organisational justice and organisational learning culture on turnover intention: A conceptual review”, Humanities and Social Sciences Reviews, 2019, 7(2), pp. 563–571

-Li, Q., Mohamed, R., Mahomed, A., & Khan, H. (2022). The Effect of Perceived Organizational Support and Employee Care on Turnover Intention and Work Engagement: A Mediated Moderation Model Using Age in the Post Pandemic Period. Sustainability, 14(15), 9125.

- Amin, M., Othman, S.Z., Saoula, O, “The Effect of Organizational Justice and Job Embeddedness on Turnover Intention in Textile Sector of Pakistan: The Mediating Role of Work Engagement” Central Asia and the Caucasus, 2021, 22(5), pp. 930–950

Methodology:

How experiment was conducted?

How participants were recruited?

What are the instructions of experiment?

How much was time given to each participant?

Others:

What are the theoretical implications of the study ?

Practical implications needs further discussion.

Add/ involve more recent citations/studies where necessary

Reviewer #2: Revised topic after correction

 The paper is generally well written and structured. However, I believe that paper has some shortcomings in terms of

Abstract: Rewrite the abstract after manuscript correction and provide picture of whole study.

Introduction

In the first paragraph of introduction used only this (2016) citation. This citation not justify the paragraph.

Introduction paragraph is not justifying the problem and bag-round of study. Revised the introduction and use the recent citations to justify and logically make connection with them.

 In the introduction (second paragraph) , the contribution of study is confused with variable relationships; why are these relationships a contribution of study? Need strong justification.

Overall, I suggest a major rewrite of the introduction. It should provide an overview of and focus on one issue with recent citations.

Literature review

Revised all literature variables and link with variables with new citation

In the literature, justify these hypotheses with literary support.

In literature, justify the conceptual model and theoretical gap.

Methodology

Where is the total population? How did you choose the sample size? And how did you choose which method, unit of analysis, and research technique to use? Provide justification. Why is this method appropriate for this data set?

General: identifying flaws in the study's design (revised methodology section) and justifying technique

Discussion

Write the theoretical contribution related to a model.

6. PLOS authors have the option to publish the peer review history of their article (what does this mean?). If published, this will include your full peer review and any attached files.

Reviewer #1: **Yes: **Oussama Saoula

Reviewer #2: **Yes: **Munwar Hussain Pahi

---

## [Author Response · Author response to Decision Letter 0]

16 Jan 2023

Point by point response to editor and reviewers

Dear editor and the reviewers,

We would like to express our profound appreciation to the editor and the reviewers for the valuable comments and suggestions made on our manuscript which were very helpful in revising and improving it.

Please note that the line numbers referred in this document is aligned with the revised manuscript which has track changes.

Reviewer 1 Comment 1: Abstract: Mention the scope of the study, the population, simple size, data collected from…

Authors’ Response: Thank you very much for the valuable comment. The suggestions have been incorporated in the revised manuscript from lines 30 to 32.

“…The study population was professionals who have been a key part of Sri Lankan startups, which involved 230 respondents. …”

Reviewer 1 Comment 2: Abstract: Mention the analysis technique/ tool used in the study

Authors’ Response: Thank you very much. Your comment is well noted. The analysis technique was added in the abstract of the revised manuscript from lines 32 to 33.

“…Data analysis was performed through a forward stepwise technique through STATA …”

Reviewer 1 Comment 3: Introduction: The introduction is not clear and very less literature is used. Follow these instructions: The introduction should briefly place the study in a broad context and highlight why it is important. It should define the purpose of the work and its significance.

Authors’ Response: Thank you very much for the detailed comment. This helps to strength the introduction with recent literatures, better argument, and justifications. The following literature were added in the introduction section with citation nos. 10, 11, 1, and 3 of the revised manuscript.

Reviewer 1 Comment 4: Introduction: The current state of the research field should be reviewed carefully, and key publications cited. Briefly mention the main aim of the work and highlight the main conclusions. Keep the introduction comprehensible to scientists working outside the topic of the paper.

Authors’ Response: Well noted your comment. A major update has done in the introduction section of the study.

Reviewer 1 Comment 5: Introduction: What is the main research focus?? Firm performance? Employee retention? Employee turnover intention?

Authors’ Response: Thank you very much for your comment. This study focuses/ aims to analyse the impact of job satisfaction, happiness, work-life balance, career management, management support, innovative work behaviour, leader-member-exchange, and co-worker support on employee turnover in startups in Sri Lanka. New paragraph has been added in the revised manuscript to explain more about firm performance, employee retention, employee turnover intention from lines 74 to 80.

“Firm performance reflects the ability of an organisation to use its human resources and other material resources to achieve its goals and objectives. Firm performance belongs to the economic category, and it should consider the use of business means efficiently during the production and consumption process [12]. Employee retention is defined as encouraging employees to remain in the organisation for a long period or the organisation’s ability to minimised employee turnover [13]. Turnover intention is the intention of the employee to change the job or organisation voluntarily [14].”

Reviewer 1 Comment 6: Introduction: Focus more on the main issue of the study

Authors’ Response: Thank you for your valuable comment. More priority was given to discuss the main issue of the study. In the first paragraph of the introduction section have updated to highlight the main issue with latest literature. New content has been added from lines 55 to 58.

“…The issue of employee turnover is considered as one of the global obstacles for organisations worldwide, which directly and adversely affects strategic plans and opportunities of gaining competitive advantages [3].…”

Reviewer 1 Comment 7: Introduction: Make the theoretical and practical gaps more clear

Authors’ Response: Thank you very much for the comment. The revised manuscript has been updated by pointing out the existing research gaps. New content has been added from lines 128 to 131.

“…to the best of the authors’ knowledge, there was no previous research done by local researchers that includes all the widely measured variables investigating the combined effect on employee turnover…”

Reviewer 1 Comment 8: Introduction: Why Sri Lanka?

Authors’ Response: Thank you for the comment. Sri Lanka has selected as the case study because to the best of the authors’ knowledge, no any previous research has been done by local researchers considering all the widely affected eight variables together. It leads to improve the introduction part of the paper. Suggestions have been incorporated in the revised manuscript from lines 127 to 130.

“…Sri Lankan context has been selected as the case study. This is because, to the best of the authors’ knowledge, there was no previous research done by local researchers that includes all the widely measured variables investigating the combined effect on employee turnover…”

Reviewer 1 Comment 9: Introduction: Why stratup s in Sri Lanka? Why employee working in startups in Sri Lanka

Authors’ Response: Thank you very much for the comment and this is well noted. Suggestions have been incorporated in the revised manuscript from lines 85 to 89. 

“…Sri Lanka has a middle rank of ease of doing business. With these favourable conditions and educational and family backgrounds, many people like to operate/apply their new idea and fill the market gap. The new generation in Sri Lanka are interested/are keen on innovations at work and being a part of unique products or services…”

Reviewer 1 Comment 10: Introduction: What was the key motivation behind focusing on factors affecting employee turnover intention in stratups in Sri Lanka?

Authors’ Response: Thank you very much for the valuable comment. The key motivation of focusing on factors affecting employee turnover intention was to gather widely affected factors together and measure the impact of each indicator at the micro level. The idea was added in revised manuscript from line 110 to 111.

“Based on their knowledge and the existing literature, authors have considered widely used factors to investigate the employee turnover issue …”

Reviewer 1 Comment 11: Introduction: Please, properly justify why the selected variables are included in the model. How did you derive the 08 variables?

Authors’ Response: Thank you for the comment. According to the past literature authors have selected widely used indicators for employee turnover and among these eight variables have been selected. The justification has included in the revised manuscript from line 110 to 117.

“Based on their knowledge and the existing literature, authors have considered widely used factors to investigate the employee turnover issue. Therefore, job satisfaction, happiness, work-life balance, career management, management support, innovative work behaviour, leader member exchange and co-worker support were selected based on previous literature findings [4-6, 8, 17-19]. As in the previous papers and along with the current study’s results, authors identified both positive and negative impacts on employee turnover among Sri Lankan startups..”

Reviewer 1 Comment 12: Introduction: As, many studies conducted in the world and in Sri Lanka about this topic, what us the main contribution of your study?

Authors’ Response: Well noted your comment. Thank you! The contribution of the study has highlighted in the revised manuscript from lines 120 to 133.

“…The present study’s scientific value can be elaborated by comparing it with previous studies. This study’s contribution can be explained in five ways. Firstly, the most critical and newly considered factors were identified together with the support of past literature. Secondly, the present study was divided/classified into different levels of employee turnover. As such, by y considering the various levels, the micro-level changes, and probabilities of the impact on employee turnover can be better identified. Further, this study helps to reduce the methodological gap. Thirdly, the Sri Lankan context has been selected as the case study. This is because, to the best of the authors’ knowledge, there was no previous research done by local researchers that includes all the widely measured variables investigating the combined effect on employee turnover. Fourthly, the analysis results can be used to identify the strengths and weaknesses of startups in Sri Lanka. Finally, this study identifies the challenges faced by startups and identifies how policy modifications can strengthen the startup ecosystem.”

Reviewer 1 Comment 13: Introduction: The paper should incorporate a more solid argumentation that allows to justify the reason that allows to select the explanatory variables that are considered in the empirical analysis.

Authors’ Response: Well noted your comment. Thank you! In the revised manuscript a paragraph was added to present the justification to select the variables in the empirical analysis from lines 110 to 117.

“Based on their knowledge and the existing literature, authors have considered widely used factors to investigate the employee turnover issue. Therefore, job satisfaction, happiness, work-life balance, career management, management support, innovative work behaviour, leader member exchange and co-worker support were selected based on previous literature findings [4-6, 8, 17-19]. As in the previous papers and along with the current study’s results, authors identified both positive and negative impacts on employee turnover among Sri Lankan startup.”

Reviewer 1 Comment 14: Literature and hypotheses development: Improve the argumentation of hypothesis. Whether the hypotheses are formulated separately or after the literature review of each section, it should be properly argued.

Authors’ Response: Thank you very much for your comment. The paper has been updated with the improved argument in literature review. The hypotheses have been formulated at the end of each sub section of literature review. New contents have incorporated as per the below line numbers.

(Line numbers 238 and 240)

“As per the literature, job satisfaction is an important factor in determining the impact on employee turnover. Accordingly, hypothesis one has been developed.”

(Line numbers from 283 to 284)

“…According to the above literature, hypothesis two has been constructed; work-life balance has a negative impact on employee turnover.”

(Line numbers from 306 to 308)

“…Based on the above cited literature, hypothesis three can be developed; employee happiness has a negative impact on employee turnover.”

(Line numbers from 336 to 337)

“…Hypothesis four has been developed based on above discussed literature.”

(Line numbers from 356 to 357)

“…Hypothesis five has been developed by concluding the above explained literature.”

(Line numbers 378 and 379)

“…With the presence of the above-mentioned literature, hypothesis six has been formulated.”

(Line numbers from 397 to 399)

“…Based on the above-mentioned literature, hypothesis seven has been developed; leader member exchange has a negative impact on employee turnover.”

(Line numbers from 417 to 419)

“…These newly published research results can be used along with all other variables that affect employee turnover. According to the above literature, hypothesis eight has been constructed.”

Reviewer 1 Comment 15: Literature and hypotheses development: Each hypothesis should be formulated at the end of a literature section of each variable presenting the different findings that have been made throughout the literature. With these arguments a reasoning should be developed in a certain direction and the conclusion of that reasoning should be the formulated hypothesis. In the current version of this manuscript the authors are including different aspects of previous literature, but it does not exist any convincing storyline in any direction.

Authors’ Response: Thank you very much for this detailed comment. The revised version has strengthened the formulation of hypothesis. Hypothesises formulations has been incorporated at the end of each sub section in the literature review and the storyline has been built. New contents have been incorporated as per the below line numbers.

(Line numbers 238 and 240)

“As per the literature, job satisfaction is an important factor in determining the impact on employee turnover. Accordingly, hypothesis one has been developed.”

(Line numbers from 283 to 284)

“…According to the above literature, hypothesis two has been constructed; work-life balance has a negative impact on employee turnover.”

(Line numbers from 306 to 308)

“…Based on the above cited literature, hypothesis three can be developed; employee happiness has a negative impact on employee turnover.”

(Line numbers from 336 to 337)

“…Hypothesis four has been developed based on above discussed literature.”

(Line numbers from 356 to 357)

“…Hypothesis five has been developed by concluding the above explained literature.”

(Line numbers 378 and 379)

“…With the presence of the above-mentioned literature, hypothesis six has been formulated.”

(Line numbers from 397 to 399)

“…Based on the above-mentioned literature, hypothesis seven has been developed; leader member exchange has a negative impact on employee turnover.”

(Line numbers from 417 to 419)

“…These newly published research results can be used along with all other variables that affect employee turnover. According to the above literature, hypothesis eight has been constructed.”

Reviewer 1 Comment 16: Literature and hypotheses development: Highlight controversial and diverging hypotheses when necessary.

Authors’ Response: Thank you for your valuable comment. This leads to build a discussion in literature review independent variables sub section. New contents have been included as per the below line numbers.

(Line numbers from 187 to 191)

“…Moreover, the descriptive statistics found a high level of job satisfaction and the intention to leave was at the mid or average level of the scale. Camara further stated that job satisfaction clearly implies the feeling about their job. But some research findings can be contradictory. Some employees are fully satisfied with the job and still want to leave the organisation for various reasons…”

(Line numbers from 205 to 207)

“…However, they didn’t observe any significant interaction between overall work-life balance and job satisfaction in predicting employee turnover intention. With these results, this indicator must be examined further.”

(Line numbers from 414 to 415)

“…It further stated that cynicism of the employee is positively associated with employee turnover…”

(Line numbers from 405 to 406)

“…However, a significantly negative impact was not evident on co-worker support.…”

Reviewer 1 Comment 17: Literature and hypotheses development: Researcher should include a summary table / review on studies conducted on Employee Turnover Intention in Sri Lanka to support the literature and arguments

Authors’ Response: Thank you very much for your comment, Literature summary table was added as an appendix, and it was cited in the revised manuscript from line numbers 168 to 169.

“Appendix S4 contains the literature summary of the above presented literature search flow diagram. The following sections present the details of each category.”

Reviewer 1 Comment 18: Literature and hypotheses development: Below papers has some interesting implications and understanding of concepts and relations that you could discuss in your introduction and literature review and how it relates to your work:

-Li, M., Li, J., Chen, X. “Employees’ Entrepreneurial Dreams and Turnover Intention to Start-Up: The Moderating Role of Job Embeddedness”, 2022, International Journal of Environmental Research and Public Health 19(15),9360

- Saoula, O.,Johari, H, “The mediating effect of organizational citizenship behavior on the relationship between perceived organizational support and turnover intention: A proposed framework” International Review of Management and Marketing, 2016, 6(7), pp. 345–354

- Saoula, O., Johari, H., Bhatti, M.A “The mediating effect of organizational citizenship behaviour on the relationship between personality traits (Big Five) and turnover intention: A proposed framework”, International Business Management, 2016, 10(20), pp. 4755–4766.

- Zito, M., Emanuel, F., Molino, M., Cortese, C. G., Ghislieri, C., & Colombo, L. (2018). Turnover intentions in a call center: The role of emotional dissonance, job resources, and job satisfaction. PloS one, 13(2), e0192126.

- Saoula, O., Johari, H., Fareed, M, “A conceptualization of the role of organisational learning culture and organisational citizenship behaviour in reducing turnover intention”, Journal of Business and Retail Management Research, 2018, 12(4), pp. 126–133

- Saoula, O., Fareed, M., Ismail, S.A., Husin, N.S., Hamid, R.A, “A conceptualization of the effect of organisational justice on turnover intention: The mediating role of organisational citizenship behaviour”, International Journal of Financial Research, 2019, 10(5), pp. 327–337.

- Poku, C. A., Alem, J. N., Poku, R. O., Osei, S. A., Amoah, E. O., & Ofei, A. M. A. (2022). Quality of work-life and turnover intentions among the Ghanaian nursing workforce: A multicentre study. PloS one, 17(9), e0272597.

- Saoula, O., Fareed, M., Hamid, R.A., Al-Rejal, H.M.E.A., Ismail, S.A, “The moderating role of job embeddedness on the effect of organisational justice and organisational learning culture on turnover intention: A conceptual review”, Humanities and Social Sciences Reviews, 2019, 7(2), pp. 563–571

-Li, Q., Mohamed, R., Mahomed, A., & Khan, H. (2022). The Effect of Perceived Organizational Support and Employee Care on Turnover Intention and Work Engagement: A Mediated Moderation Model Using Age in the Post Pandemic Period. Sustainability, 14(15), 9125.

- Amin, M., Othman, S.Z., Saoula, O, “The Effect of Organizational Justice and Job Embeddedness on Turnover Intention in Textile Sector of Pakistan: The Mediating Role of Work Engagement” Central Asia and the Caucasus, 2021, 22(5), pp. 930–950

Authors’ Response: Thank you very much for the detailed comment and sharing the latest literature related to this paper. New literature has been incorporated in the introduction, literature review and results and discussion sections the paper as per the below line numbers.

(Line numbers from 45 to 46)

“…Companies need to give high priority to employee development and predict employee behaviour [1]…”

(Line numbers from 55 to 58)

“…The issue of employee turnover is considered as one of the global obstacles for organisations worldwide, which directly and adversely affects strategic plans and opportunities of gaining competitive advantages [3]…”

(Line numbers from 66 to 68)

“…Further, promoting employee well-being leads to decrease employee turnover [10]. Providing psychological and social support through counselling promotes the quality of work-life [11]...”

(Line numbers from 318 to 325)

“A cross-sectional survey has been conducted for front-line healthcare staff in China by Li, Mohamed [30] to measure the impact of organisational support on employee turnover intention. This study’s results could verify that organisational support negatively affected employee turnover intention. Saoula and Johari [31] studied this area and determined a negative relationship between organisational support and employee turnover intention. As both of the above explained research have been conducted in non-Western countries, the findings help to complete the theoretical framework for the current study in the Sri Lankan context.”

(Line numbers from 344 to 349)

“Saoula and Johari [31] researched the effect of personality traits (big five) on employee turnover intention. The researchers state that the relationship between the big five personality traits and turnover intention will support early prediction of employee turnover intentions. Identifying employee’s personalities and helping them to find the most suitable job role is a long-term process, though it will be highly advantageous for both employees and the organisation.”

(Line numbers from 365 to 373)

“The organisational learning culture is a key factor for innovative work behaviour. Saoula, Fareed [36] conducted research in Malaysia, a developing country in Asia to examine the relationship between organisational learning culture and employee turnover intention. The organisational learning culture improves learning capability, supports sustainable development, and affects organisation's positive changes. As organisational learning culture and employee turnover intention have a negative relationship, the result helps to identify the impact of innovative work behaviour. According to the existing/available literature, limited studies have been conducted on this topic.”

(Line numbers from 608 to 612)

“Entrepreneurs are the founders of startups. Employees’ entrepreneurial dreams positively affect employee intention to startups. Employees in the startups also will have an ideation to start their own business. According to the study by Li, Li [44] the mediating role of employees’ entrepreneurial self-efficacy and the moderating role of job embeddedness in the influence of entrepreneurial dreams on employees’ turnover intention to startup.”

Reviewer 1 Comment 19: Methodology: How experiment was conducted?

Authors’ Response: Thank you for the comment. The flow of methodology could improve with the help of the next three comments, including this. The experiment was conducted using both online and manual channels. 

Accordingly, the revised manuscript is updated as follows in lines 432 and 435.

“…The authors directly distributed the questionnaire. Moreover, authors could contact management in startups and distribute the questionnaire in their organisation.…”

Reviewer 1 Comment 20: Methodology: How participants were recruited?

Authors’ Response: Duly noted with thanks! The participants were selected by random sampling method. Authors could contact the management of respective organisations to reach the respondents.

The methodology part has been written in descriptive manner in the revised document. From lines 432 to 435 and lines 457 to 458 were newly added.

“…The authors directly distributed the questionnaire. Moreover, authors could contact management in startups and distribute the questionnaire in their organisation….”

“…The researchers applied a random sampling method, mainly employees who are a part of or have been a part of the startup …”

Reviewer 1 Comment 21: Methodology: What are the instructions of experiment?

Authors’ Response: Thank you very much for the comment and well noted. 

Instructions for the experiments were.

• Participants should be a part of the startup 

• He/she should consider the behaviour and culture of that startup when answering the questions

• Respondent should answer all the questions

The instructions given in the questionnaire has included in the revised manuscript from lines 444 to 449.

“…A minimum of four questions was added under each indicator. The researchers facilitated anonymously answering all the questions in the questionnaire. The participants should be a part of startup and he/she should consider the behaviour and culture of that startup when answering the questions. All nine indicators were covered by Likert scale questions from 1 to 5 rating scale, depicting (1) strongly disagree to (5) strongly agree to collect respondents’ attitudes and opinions…”

Reviewer 1 Comment 22: Methodology: How much was time given to each participant?

Authors’ Response: Thank you very much for the comments. This helps to build the story line in methodology part. 15 minutes time were given to answer the questionnaire. New content has added from lines 449 to 452

“…Each respondent took about 10-15 minutes to complete answering the questionnaire and took approximately 5-7 minutes to fill out the questionnaire…”

Reviewer 1 Comment 23: What are the theoretical implications of the study?

Authors’ Response: Well noted your comment. Thank you very much! In revised manuscript has added new sub section to discuss theoretical implications from lines 648 to 654.

“The current study empirically investigated the impact of job satisfaction, innovative work behaviour, co-worker support and leader member exchange on employee turnover. According to the authors’ knowledge, no prior studies were conducted considering the combined impact of all the independent variables on employee turnover. Therefore, this study strengthens the literature by demonstrating how job satisfaction, innovative work behaviour, co-worker support and leader member exchange impact employee turnover in Sri Lankan startups. 

 The findings reveal that job satisfaction has a negative impact on employee turnover. This finding is consistent with the previous study, job satisfaction significantly predicted employee turnover [6]. This study consolidates past findings that male employees have higher turnover intention than female employees. Female employees have comparatively higher-level job satisfaction [8]. This study implies that employees age 31 to 40 years have high employee turnover intention. The research findings are similar to Lu, Lu [8]; the older employees have high intentions to leave the company.”

Reviewer 1 Comment 24: Practical implications need further discussion.

Authors’ Response: Thank you very much for your valuable comment. Practical implications section was improved with further discussion. New contents have been added from lines 665 to 671, 674 to 680, 686 to 690.

“…This study provides managerial insights on lowering employee turnover in Sri Lankan startups. First, startups need to be aware that experienced employees in startups can be easily taken by well-established companies because, later, they have hand on experience and skills. Therefore, it is important to implement strategies for a solid career development plan, career growth, personal status, and employee recognition. As job satisfaction can predict employee turnover, it is a must to measure those indicators and maintain a favourable level at all times.”

“…More importantly, healthy LMX can boost employees’ work engagement. This healthy level can maintain by conducting regular meetings, training programs and informal mentorship with employees’ immediate supervisors [8]. Further, management can allow employees at all levels to present their fresh ideas and incorporate them to influence organisation’s decision-making process. These processes can lower employee hierarchy and build strong relationships while recognising them in the company.”

“…Furthermore, having a flexible approach to work in an organisation culture will increase the trust between employees and the organisation. Giving the freedom to take risks and not allowing them to feel alone during work will give value to employees. Finally, all the above actions will strongly impact reducing employee intention to leave the organisation.”

Reviewer 1 Comment 25: Add/ involve more recent citations/studies where necessary

Authors’ Response: Thank you very much for the comment and this is well noted. New citations were added in revised manuscript in introduction, literature review and results and discussions sections as per the below line numbers.

(Line numbers from 45 to 46)

“…Companies need to give high priority to employee development and predict employee behaviour [1]…”

(Line numbers from 55 to 58)

“…The issue of employee turnover is considered as one of the global obstacles for organisations worldwide, which directly and adversely affects strategic plans and opportunities of gaining competitive advantages [3]…”

(Line numbers from 66 to 68)

“…Further, promoting employee well-being leads to decrease employee turnover [10]. Providing psychological and social support through counselling promotes the quality of work-life [11]...”

(Line numbers from 318 to 325)

“A cross-sectional survey has been conducted for front-line healthcare staff in China by Li, Mohamed [30] to measure the impact of organisational support on employee turnover intention. This study’s results could verify that organisational support negatively affected employee turnover intention. Saoula and Johari [31] studied this area and determined a negative relationship between organisational support and employee turnover intention. As both of the above explained research have been conducted in non-Western countries, the findings help to complete the theoretical framework for the current study in the Sri Lankan context.”

(Line numbers from 344 to 349)

“Saoula and Johari [31] researched the effect of personality traits (big five) on employee turnover intention. The researchers state that the relationship between the big five personality traits and turnover intention will support early prediction of employee turnover intentions. Identifying employee’s personalities and helping them to find the most suitable job role is a long-term process, though it will be highly advantageous for both employees and the organisation.”

(Line numbers from 365 to 373)

“The organisational learning culture is a key factor for innovative work behaviour. Saoula, Fareed [36] conducted research in Malaysia, a developing country in Asia to examine the relationship between organisational learning culture and employee turnover intention. The organisational learning culture improves learning capability, supports sustainable development, and affects organisation's positive changes. As organisational learning culture and employee turnover intention have a negative relationship, the result helps to identify the impact of innovative work behaviour. According to the existing/available literature, limited studies have been conducted on this topic.”

(Line numbers from 612 to 616)

“Entrepreneurs are the founders of startups. Employees’ entrepreneurial dreams positively affect employee intention to startups. Employees in the startups also will have an ideation to start their own business. According to the study by Li, Li [44] the mediating role of employees’ entrepreneurial self-efficacy and the moderating role of job embeddedness in the influence of entrepreneurial dreams on employees’ turnover intention to startup.”

Reviewer 2 Comment 1: Abstract: Rewrite the abstract after manuscript correction and provide picture of whole study.

Authors’ Response: Thank you very much for your comment. Abstract has been rewritten after doing the manuscript corrections.

Reviewer 2 Comment 2: In the first paragraph of introduction used only this (2016) citation. This citation does not justify the paragraph.

Authors’ Response: Thank you very much for the comment. The 1st paragraph of introduction section has upgraded with recent literatures with the citation nos. 1, and 3. 

New contents have included from lines 45 to 46, and lines 55 to 58.

“…Companies need to give high priority to employee development and predict employee behaviour [1]…”

“…The issue of employee turnover is considered as one of the global obstacles for organisations worldwide, which directly and adversely affects strategic plans and opportunities of gaining competitive advantages [3]…”

Reviewer 2 Comment 3: Introduction paragraph is not justifying the problem and bag-round of study. Revised the introduction and use the recent citations to justify and logically make connection with them.

Authors’ Response: Thank you so much for the comment. The introduction section was upgraded with recent literatures with the citation nos. 10, 11, 1, and 3 and justifications.

Reviewer 2 Comment 4: In the introduction (second paragraph), the contribution of study is confused with variable relationships; why are these relationships a contribution of study? Need strong justification.

Authors’ Response: Thank you very much for your valuable comment. The content has been updated with recent citations in line 64.

“Many variables influence employee turnover intentions [4-6]…”

Reviewer 2 Comment 5: Overall, I suggest a major rewrite of the introduction. It should provide an overview of and focus on one issue with recent citations.

Authors’ Response: Thank you very much, the comment well noted. New literatures have added in revised manuscript and highlighted the main issues and the research gaps. Every paragraph of the introduction has updated according to the reviewers’ comments.

Reviewer 2 Comment 6: Revised all literature variables and link with variables with new citation.

Authors’ Response: Thank you very much for the comment and well noted. After adding new literatures, revised all literature variables and linked with variables with new citation.

Reviewer 2 Comment 7: In the literature, justify these hypotheses with literary support.

Authors’ Response: Thank you very much for the comment. A storyline was developed on hypothesis formulation. New contents have been incorporated as per the below line numbers.

(Line numbers 238 and 240)

“As per the literature, job satisfaction is an important factor in determining the impact on employee turnover. Accordingly, hypothesis one has been developed.”

(Line numbers from 283 to 284)

“…According to the above literature, hypothesis two has been constructed; work-life balance has a negative impact on employee turnover.”

(Line numbers from 306 to 308)

“…Based on the above cited literature, hypothesis three can be developed; employee happiness has a negative impact on employee turnover.”

(Line numbers from 336 to 337)

“…Hypothesis four has been developed based on above discussed literature.”

(Line numbers from 356 to 357)

“…Hypothesis five has been developed by concluding the above explained literature.”

(Line numbers 378 and 379)

“…With the presence of the above-mentioned literature, hypothesis six has been formulated.”

(Line numbers from 397 to 399)

“…Based on the above-mentioned literature, hypothesis seven has been developed; leader member exchange has a negative impact on employee turnover.”

(Line numbers from 417 to 419)

“…These newly published research results can be used along with all other variables that affect employee turnover. According to the above literature, hypothesis eight has been constructed.”

Reviewer 2 Comment 8: In literature, justify the conceptual model and theoretical gap.

Authors’ Response: Well noted your comment. Thank you! Conceptual model and theoretical gap have justified in the revised manuscript from lines 420 to 424.

“These studies have a common limitation in gathering more independent variables and analysing the impact. Therefore, a need exists to measure the effect of job satisfaction, work-life balance, happiness, management support, career management, innovative work behaviour, leader member exchange, and co-worker support together on employee turnover.”

Reviewer 2 Comment 9: Where is the total population? How did you choose the sample size? And how did you choose which method, unit of analysis, and research technique to use? Provide justification. Why is this method appropriate for this data set?

Authors’ Response: Thank you very much for this valuable comment.

Total population was 1300 and sample size identified by referring calculator.net online sample size calculator. A stepwise ordered probit analysis method was used as the selected variables are widely used indicators for employee turnover therefore authors required to do a micro level analysis for these variables. 

The details of sampling have added in revised manuscript from lines 459 to 465.

“…The sample size was selected by referencing the Krejcie and Morgan sampling table and Calculator.net [40] with a confidence level of 95% and 7% of margin of error. The calculation results indicated a minimum of 171 professionals. A stepwise ordered probit analysis method was used as the selected variables are widely used indicators for employee turnover; therefore, a micro-level analysis was required to study how these variables impact. A pilot survey was conducted to identify whether the purpose of the questions was clear to the respondents.”

Reviewer 2 Comment 10: General: Identifying flaws in the study's design (revised methodology section) and justifying technique.

Authors’ Response: Well noted your comment. Thank you! Methodology section has been updated in revised manuscript from line 520 to 522.

“…The probit model is an estimation technique for equations with dummy dependent variables that avoids the unboundedness problem of the linear probability model by using a variant of the cumulative normal distribution [42]…”

Reviewer 2 Comment 11: Discussion: Write the theoretical contribution related to a model.

Authors’ Response: Thank you very much for your valuable comment.

The revised manuscript has been updated with a new sub section to discuss theoretical implications from lines 648 to 661.

“The current study empirically investigated the impact of job satisfaction, innovative work behaviour, co-worker support and leader member exchange on employee turnover. According to the authors’ knowledge, no prior studies were conducted considering the combined impact of all the independent variables on employee turnover. Therefore, this study strengthens the literature by demonstrating how job satisfaction, innovative work behaviour, co-worker support and leader member exchange impact employee turnover in Sri Lankan startups. 

 The findings reveal that job satisfaction has a negative impact on employee turnover. This finding is consistent with the previous study, job satisfaction significantly predicted employee turnover [6]. This study consolidates past findings that male employees have higher turnover intention than female employees. Female employees have comparatively higher-level job satisfaction [8]. This study implies that employees age 31 to 40 years have high employee turnover intention. The research findings are similar to Lu, Lu [8]; the older employees have high intentions to leave the company.”

Reviewer 2 Comment 12: Revised topic after correction

Authors’ Response: Well noted your comment. The topic has been updated in lines 1 to 2 and 19 to 20, and the new topic is,

“Factors impacting employee turnover intentions among professionals in Sri Lankan startups”

---

## [Decision Letter · Decision Letter 1]

31 Jan 2023

Factors impacting employee turnover intentions among professionals in Sri Lankan startups

PONE-D-22-30684R1

Dear Dr. Lakshmi Kanchana,

We’re pleased to inform you that your manuscript has been judged scientifically suitable for publication and will be formally accepted for publication once it meets all outstanding technical requirements.

Kind regards,

Muhammad Fareed, Ph.D

Academic Editor

PLOS ONE

Additional Editor Comments (optional):

Dear Author/s,

Thank you for making all the corrections.

Reviewers' comments:

Reviewer's Responses to Questions

**Comments to the Author**

1. If the authors have adequately addressed your comments raised in a previous round of review and you feel that this manuscript is now acceptable for publication, you may indicate that here to bypass the “Comments to the Author” section, enter your conflict of interest statement in the “Confidential to Editor” section, and submit your "Accept" recommendation.

Reviewer #1: All comments have been addressed

Reviewer #2: All comments have been addressed

2. Is the manuscript technically sound, and do the data support the conclusions?

Reviewer #1: Yes

Reviewer #2: (No Response)

3. Has the statistical analysis been performed appropriately and rigorously? 

Reviewer #1: Yes

Reviewer #2: Yes

4. Have the authors made all data underlying the findings in their manuscript fully available?

Reviewer #1: Yes

Reviewer #2: Yes

5. Is the manuscript presented in an intelligible fashion and written in standard English?

Reviewer #1: Yes

Reviewer #2: Yes

6. Review Comments to the Author

Reviewer #1: The authors have adequately addressed the comments raised in a previous round of review and I feel that this manuscript is now acceptable for publication

Reviewer #2: (No Response)

7. PLOS authors have the option to publish the peer review history of their article (what does this mean?). If published, this will include your full peer review and any attached files.

Reviewer #1: **Yes: **Oussama Saoula

Reviewer #2: No

---

## [Editor Report · Acceptance letter]

1 Feb 2023

PONE-D-22-30684R1 

Factors impacting employee turnover intentions among professionals in Sri Lankan startups 

Dear Dr. Jayathilaka:

I'm pleased to inform you that your manuscript has been deemed suitable for publication in PLOS ONE. Congratulations! Your manuscript is now with our production department. 

Kind regards, 

on behalf of

Dr. Muhammad Fareed 

Academic Editor

PLOS ONE